# Perspectives of Hospital Staff on Barriers to Smoking Cessation Interventions among Drug-Resistant Tuberculosis Patients in a South African Management Hospital

**DOI:** 10.3390/ijerph21091137

**Published:** 2024-08-28

**Authors:** Phindile Zifikile Shangase, Nduduzo Msizi Shandu

**Affiliations:** 1Division of Public Health, Faculty of Health Sciences, University of Free State, Bloemfontein 9300, South Africa; shangasezpc@ufs.ac.za; 2Department of Human Movement Science, Faculty of Science and Agriculture, University of Zululand, KwaDlangezwa 3886, South Africa

**Keywords:** drug-resistant TB, cigarette smoking, smoking cessation, respiratory diseases, tuberculosis, hospital staff

## Abstract

Drug-resistant tuberculosis (DR-TB) remains a major cause of illness and death, with personal and non-addiction-related barriers. This study aimed to explore the perspectives of hospital staff on barriers to smoking cessation interventions (SCIs) for in-patients at a DR-TB management hospital in Durban, KwaZulu-Natal, South Africa. In-depth interviews were conducted with a purposive sample of eighteen hospital staff (HS), and the data were analyzed using NVivo 10. Three core themes were identified: patients’ barriers (addiction to tobacco, relapse after improvement in health, and non-disclosure of smoking status to HS), staff personal barriers (poor knowledge of smoking’s effect on treatment outcomes and smoking cessation aids), and institutional barriers (staff shortage, time constraints, lack of pharmacological smoking cessation aids, access to cigarettes around hospital premises, and SCIs not prioritized and not assigned to a specific category of HS). Training on SCIs for HS, assigning SCIs to specific HS, integrating SCIs within existing services, and banning access to cigarettes within the hospital premises are assumed to assist DR-TB patients in smoking cessation, improving their response to TB treatment and overall health outcomes.

## 1. Introduction

The World Health Organization’s (WHO) Global Tuberculosis (TB) Report 2023 details comprehensive and up-to-date statistics of the disease epidemic and of success in prevention, diagnosis, and treatments at the global, country, and regional levels [1]. Along these lines, the 2023 edition of the report indicates that approximately 192 countries and areas (out of 215) with more than 99% of the world’s population and TB cases reported data [1]. South Africa (SA) is ranked among these countries, with the highest TB and multi-drug-resistant TB (MDR-TB) burden, with incidence rates of 781 and 34 cases, respectively, per 100,000 [2]. Drug-resistant TB comprises multi-drug-resistant (MDR) and extensively drug-resistant (XDR) TB diseases, which are challenging to treat. MDR-TB is resistant to at least isoniazid and rifampicin, while XDR-TB is resistant to any fluoroquinolone and at least one of three injectable second-line TB drugs [1,3]. 

Regrettably, the significant global incidence of tuberculosis has been further exacerbated by the rise in tobacco smoking, resulting in a catastrophic synergistic relationship, especially in SA [4]. An analysis study on the SA’s 2016 Demographic and Health Survey shows that the overall tobacco smoking prevalence values in men and women in the country are 11% and 7%, respectively, with one in five being multimorbid [5]. Smoking is a costlier health risk for DR-TB patients, whose health outcomes are more adversely affected by cigarettes [6]. 

Tobacco use is a leading cause of respiratory infections adversely affecting the lung immune cell functions [4] and reducing the efficacy of TB treatment [7,8]. Jeyashree et al. [9] reported the reversal of most smoking-related immunologic abnormalities within six weeks of stopping smoking; consequently, smoking cessation interventions (SCIs) are recommended for positive health outcomes and the successful treatment of TB patients. Possible adverse chemical reactions between anti-TB drugs and substances contained in tobacco, including nicotine, may reduce TB treatment outcomes as noted in some experimental studies [10,11,12,13,14,15]. 

The WHO estimated that deaths from tobacco-related illnesses will amount to more than eight million people each year by 2030 [16] and strongly recommended the registration of TB patients who smoke to enable counselling and the provision of appropriate treatment [17]. A mathematical modeling analysis estimated that, in TB management, decreasing the smoking prevalence by 1% annually until complete cessation would prevent 27 million smoking-attributable deaths from TB by 2050 [18]. In SA, smoking has been reported as one of the co-exposures and possible reasons for the poorer survival of DR-TB patients in a retrospective cohort study that included participants within the KwaZulu-Natal (KZN) province [19,20]. Previous studies conducted in South African TB clinics identified high current smoking rates among TB patients, 21.8% by Louwagie and Ayo-Yusuf and 20% by Peltzer et al. [21,22]. 

Hospital staff (HS) face various challenges in providing care for TB patients who smoke as part of TB treatment programs. These challenges include staff workload, lack of resources and motivation [23,24], and lack of adequate training among physicians on cessation methods, such as the importance of the tradition of physical activity [25]. TB patients tend to be more receptive to tobacco cessation messages due to the effect of smoking on their health outcomes [26] than general smokers with low receptivity to tobacco cessation messages [27]. In SA, there are published guidelines for treating tobacco use among TB patients [28], including evidence of their efficacy in clinical trials [29], but no SCIs have been reported in South African hospitals [30,31]. However, HS can influence smoking cessation among TB patients due to their close and frequent involvement with them [32] as TB disease provides teachable moments for smoking cessation [33], especially among hospitalized patients [34]. In this regard, the rationale of this study is to address the critical gap in the implementation of smoking cessation interventions for TB patients in South African hospitals. Given the exacerbating effects of smoking on TB treatment outcomes, there is an urgent need to understand the perspectives of hospital staff and identify the barriers they face in providing effective smoking cessation support. On that note, the study aims to enhance TB treatment programs and improve health outcomes for TB patients who smoke.

Although the South African government has implemented comprehensive tobacco control policies since 1993 [35], such policies are deemed more effective in reducing smoking prevalence with supportive services in place [36]. The South African National Department of Health’s guidelines on the management of people with TB in special circumstances recommend behavioral and pharmacological treatment interventions for tobacco dependence to be applied by healthcare professionals [28]. However, this is not achieved in SA and consequently leads to a continuously increasing smoking prevalence among MDR-TB patients and direct exposure to premature morbidity and mortality, which negatively impacts their life expectancies [4,37]. Therefore, this study aims to explore HS’s perspectives concerning SC support for DR-TB in-patients and existing barriers to SCIs to strengthen the SC aspect of TB programs in SA.

## 2. Materials and Methods

### 2.1. Study Participants and Ethical Approval 

The sample included eighteen (N = 18) hospital staff (HS) who care for in-patients at the DR-TB hospital. HS who worked at the outpatient clinic were excluded from the study. Table 1 presents the breakdown of the study participants. The participants were invited via posters, emails, and word-of-mouth. The study utilized a purposive sampling method in which the inclusion criteria required the participants to be directly involved in the care of in-patients with drug-resistant tuberculosis (DR-TB), ensuring they had firsthand experience of the challenges and dynamics of providing smoking cessation support in a hospital setting. The participants were required to have a minimum of one year of experience working in the hospital to ensure familiarity with the institutional protocols and patient demographics. The University of KwaZulu-Natal’s Biomedical Research Committee provided ethical approval for this study (Reference No: BE 493/14). Permission was also obtained from the KZN Department of Health and the hospital management. HS consented individually after being informed about the study and their right to withdraw at any point if they so desired. The participants were assured of confidentiality, and code names were used to identify them to ensure anonymity. All participants voluntarily agreed to participate in the study.

### 2.2. Study Design and Setting

A qualitative research design, which falls within the constructivist paradigm [38], was used to investigate the HS’s perspectives about SC support for DR-TB patients. Smoking cessation practices of HS were also explored. The study was conducted at the DR-TB management hospital in Durban, KZN province. KZN’s population is the second largest in South Africa, estimated at 11.1 million (19.9% of the total national population) [39,40]. KZN also recorded the second highest incidence rate of TB in 2015, 685 per 100,000 [40], and an XDR-TB incidence of 3.5 cases/100,000 (776 cases) in 2011–2012. The majority of the districts in KZN are experiencing a rise in the incidence of TB [41]. 

### 2.3. Data Collection

In-depth interviews were conducted with a purposive sample of eighteen HS between September and October 2015. Each interview lasted for an average of 30 min. Open-ended questions were asked on the availability of resources for SC support, barriers to SC support, and knowledge of the risks of smoking on DR-TB. New questions were asked to follow a line of inquiry introduced by the participant. Interviews were conducted in the English language at the hospital premises, tape-recorded with both the verbal and written consent of the participants and then transcribed verbatim. 

### 2.4. Data Analysis

A thematic analysis was initially carried out on the data by the first author, while the supervisor and research assistant provided the objective of the evaluation of the resulting themes. This was achieved through the utilization of an inductive method to code the data collected. The software NVivo 10 (Lumivero, Burlington, MA, USA) was used to aid the data analysis process [42]. 

## 3. Results

The sample included eighteen [N = 18] HS who care for in-patients at the DR-TB hospital, which included doctors, nurses, pharmacists, and social workers. HS who worked at the outpatient clinic were excluded from the study. Table 1 below presents the breakdown of the study participants. Three core themes were identified (barriers to smoking cessation), and subthemes under each core theme were delineated, as presented in Table 2. Personal factors (patients) are smoking cessation barriers within the individual DR-TB inpatient, while personal factors (staff) are barriers to smoking cessation that lie with hospital staff. Structural (institutional) barriers to smoking cessation are factors that are the responsibility of hospital management and the government. The participants also suggested ways to address SC among DR-TB in-patients. 

### 3.1. Personal Factors (Patients) 

(i)Non-disclosure of smoking status by patients

The participants were concerned that in-patients do not disclose their smoking status when asked at the point of admission into the health facility. They opined that the non-disclosure of in-patients’ smoking status is one of the reasons SC is never addressed properly as part of DR-TB treatment in the health facility. 


*“…. Out on the ground you will find cigarette butts, we can’t pinpoint who is smoking because some of them will do it secretly and the only way for us to know that the patients are smoking is to smell the cigarette, but if we go out to address them we can’t find anyone”*
(Participant 7; Female, Nurse).

The reasons cited for patients’ non-disclosure of their smoking status include worry that they would be denied treatment or held responsible for their initial treatment failure. 


*“…they think it [disclosing their smoking status] will decrease the chances of them being admitted, unaware that they will be admitted even if they smoke. Though we do advise them that they ought to decrease or stop smoking”*
(Participant 9; Male, Nurse).

Due to the tendency for patients to hide their smoking status, hospital staff devised other means of investigating patients’ smoking status. Some of these ways included the persistent probing of the patient and asking the patient’s relatives, friends, and other in-patients.


*“We make the follow-up to check if the patient does continue to smoke. For example, we ask relatives and friends because they usually hide from those watching them”*
(Participant 1; Female, Nurse).

One participant expressed the need for means to determine smoking status since in-patients tend to provide false information when asked.


*“…yeah, and whether they tell us the truth or not, we don’t know because there is no other way for us to detect, if they smoke or not. If there can be something we will use it, just as we do to determine the patients who are diabetic…”*
(Participant 7; Female, Nurse).

(ii)Relapse due to improved TB conditions

Generally, the participants reported that improved DR-TB conditions led smokers to resume smoking, especially under the influence of their friends as related in the following narrative.


*“…There are friends that a person leaves when they are admitted, we advise the friends not to bring cigarettes, and even the patients won’t smoke because they are still ill. But once they have recovered, the friends entertain him/her with cigarettes, and he/she will start smoking again….so it’s a vicious cycle”*
(Participant 4; Female, Nurse).

The following participant believed that the “pass-out” (visiting home for a few days) that is authorized upon improvements in DR-TB conditions enables in-patients to engage in smoking while away.


*“Someone would take “pass-out” and go to town to smoke, or say he/she is going home for a week, they do everything without our knowledge, so when he/she comes back their health condition is worse”*
(Participant 4; Female, Nurse).

HS also explained that DR-TB patients who smoke sometimes do so deliberately so that they can be sent home, especially when they witness some improvements in their health condition.


*“He/she is going to be shouted at by the Doctor [if found smoking] and the Doctor tells him/her that he/she is going to be discharged. But some patients can just be aggressive and say ‘Let me go now’…some are defensive, more especially when they see that they are much better. So they just do wrong intentionally”*
(Participant 4; Female, Nurse).

(iii)Addiction

The addictive property of cigarettes was reported by most participants as the main reason for DR-TB patients failing to quit smoking. 


*“It is hard, but we usually confront them, because they used to hide at the back of the container, they cannot just stop because they are addicted but they will find another place where they can smoke”*
(Participant 16; female, Social Worker).

### 3.2. Personal Factors (Staff)

(i)Lack of knowledge regarding smoking cessation aids

HS in this study were not very conversant with smoking cessation aids, like nicotine replacement therapy (NRT). 


*“…I do not know much about smoking aids, but some can be used such as the smoker’s gum”*
(Participant 9; Male, Nurse).

(ii)Knowledge about the effects of smoking on TB recovery 

HS expressed varied knowledge regarding the effects of smoking on TB. Their responses differed according to their experiences and professions, as described in the following narrative.


*“…smoking damages the bronchial tubes. That’s my belief, and so if you are a smoker and you continue to smoke probably your response to treatment will not be as good as it can be, and it will take longer for the smoker to heal. Probably it allows TB to get to you more easily if you are a smoker because your protective mechanism is damaged”*
(Participant 11; Male, Doctor).

A nurse participant described the effects of nicotine on the blood vessels, hence affecting the effectiveness of TB treatment.


*“Yes, it does affect their [TB patients’] recovery; it delays the healing process when you smoke. Cigarettes have something called nicotine, nicotine affects the blood vessels. So, when they smoke, they are blocking the effects of the medication”*
(Participant 3; Male, Nurse).

One participant displayed poor knowledge about the addictiveness and effects of cigarette smoking on health.


*“…maybe no one will even get to the serious stage of illness due to smoking…because it is not a disease whereby the doctor can prescribe this or whatever aids that can be used. Here there is no one who we can say cannot live without a cigarette”*
(Participant 4; Female, Nurse).

### 3.3. Structural (Institutional Factors)

(i)Lack of smoking cessation interventions

The only form of SCIs mentioned by the participants was general health education. Some participants mentioned that no SCI existed in the hospital for DR-TB in-patients who smoke. Additionally, a social worker reiterated that they normally offer talks on smoking around the hospital wards during the health education sessions, but voiced concerns at the lack of information pamphlets to aid their efforts at educating the patients about the hazards of smoking. Occupational therapy was also mentioned as a way of keeping the patients engaged while they are admitted.


*“We don’t have any formal program except that we keep them busy by taking them to occupational therapy where they do things like beadwork, cooking lessons and things like that, just to keep them contained”*
(Participant 3; Male, Nurse).

When asked if patients could be willing to use SCIs if available within the hospital, one participant mentioned that, due to their lack of knowledge on the effects of smoking on their health, not all participants would want SCIs.


*“…not all of them, because in most cases patients who are admitted to government hospitals are people who occupy low status within the community, so they are not well educated. So if they refused to do something they won’t do it, they are not well equipped with knowledge”*
(Participant 5; Male, Nurse).

(ii)Emphasis on other health education matters

DR-TB in-patients were provided daily health pep talks. The participants mentioned that these pep talks usually covered general topics on health education. SC issues were only sometimes presented as part of some of these pep talks, as explained in the following narrative.


*“The main emphasis is on washing hands and taking the pills the right way. However, different topics are covered (e.g., on smoking) and they are allowed to present on any health issue that might be of importance on that day.”*
(Participant 1; Female, Nurse).

However, the participants explained that the health education programs that are offered in the hospital wards were ineffective because smoking seemed to be a less concerning issue to the in-patients than their current state of health. 


*“…no there is no way for follow-up; I don’t want to lie in that case. We usually host health education and we inform them everything important as health professionals, and we tell them that they should change their lifestyle because it’s dangerous.”*
(Participant 5; Male, Nurse).

(iii)Smoking cessation aids not considered in the government’s list of essential medicines

HS mentioned that, for SC aids to be available to patients in government hospitals, such aids must be included in the government’s list of essential medicines, and SC aids are not considered essential.


*“In the public sector, we don’t keep any nicotine replacement therapies…we’re working with limited resources so sometimes we don’t have sufficient finances to procure all the drugs that we need and although smoking cessation is a big part of the patients care it will all depend on the cost-effectiveness analysis of whether nicotine replacement patch should become part of the list of essential medicines as it is not classified as a medicine”*
(Participant 15; Female, Pharmacist).

(iv)Priority on treatment adherence only and lack of dedicated staff for SCIs

Due to the adverse health situation that DR-TB in-patients are experiencing, HS tends to only focus on what is directly related to the in-patient’s state of health concerning the DR-TB infection. The participants mentioned that the usual health focus is on addressing issues like adherence to treatment regimen alone, rather than including SC interventions for in-patients.


*“It’s treatment adherence that’s what we emphasize on and counsels them on how they should take their medication”*
(Participant 3; Male, Nurse).

HS also reported that SC is not included in the job description of any of the staff; hence, it is not prioritized. 


*“…nobody has it (smoking cessation) as part of their job description, to counsel the patient of smoking cessation”*
(Participant 15; Female, Pharmacist).

(v)Access to cigarettes around hospital premises and non-compliance to smoking restrictions 

The participants explained that vendors, HS, relatives, and friends help in-patients to access cigarettes. 


*“There are vendors on the other side of our fences and we can’t stop them. They also ask other patients to buy for them, some ask some of the staff to buy for them also”*
(Participant 3; Male, Nurse).

It was also reported that in-patients sometimes shared cigarettes with their peers while in the hospital.


*“They smoke together…. They spend lots of time together, thus they develop friendship, (you understand) they will bond.”*
(Participant 5; Male, Nurse).

(vi)Lack of time and staff shortage

Most participants (89%; n = 16) explained that they do not have enough time (to carry out extra interventions like SC) due to various ward duties and limited staff.


*“After all, it may be difficult to determine the exact time I spend with the patient because they have different problems. It is not enough because we do not have sufficient staff members and there is too much work. I must say that we are busy all day and we work tirelessly when attending to the queries of the patients”*
(Participant 1; Female, Nurse).

HS reported a lack of time to carry out one-on-one consultations with in-patients.


*“…. It is very difficult because we have to give attention to two or more patients at the same time so one-on-one time is a big no no, we cannot do that…not unless the patient needs special attention, you would then attend to that person as an individual”*
(Participant 2; Female, Nurse).

Other participants also expressed concern over staff shortages. In the opinion of these participants, the shortage of staff is one of the issues hindering the introduction of SCIs to in-patients.


*“…but the main part of the hospital is so busy, so under-staffed, such large number of patients! So I don’t think it (smoking cessation intervention) will be a priority program in the near future. It will take a long time to come…because the staff is not enough”*
(Participant 11; Male, Doctor).

### 3.4. Suggestions on How to Address Smoking among TB Patients

The participants suggested the creation of a dedicated smoking cessation facility for SC where in-patients who smoke could be referred to access interventions.


*“I think it would be good to have a facility that we can refer them to where they can get help…”*
(Participant 3: Male, Nurse).

The participants also recommended the integration of SCIs into already existing programs as a way to address smoking among DR-TB in-patients.


*“…I think it can be a part of the public health awareness and other awareness campaign, it could be put as a part of TB education or HIV education, you could slip it there maybe not as a separate program but part of other programs”*
(Participant 11; Male, Doctor).

## 4. Discussion

HS mentioned various factors serving as barriers to SCIs for DR-TB in-patients. Some of these are personal factors relating to the patients as well as hospital staff, and factors which the hospital management and government are responsible for. 

### 4.1. Personal Factors (In-Patients)

In this study, personal factors were identified relating to in-patients, such as non-disclosure of patient’s smoking status, addiction to nicotine, and the tendency for patients who stop smoking while they are very sick to resume when their health improves. Louwagie et al. [29] found in their SA study that a TB diagnosis serves as motivation to consider SC, while improved TB conditions lead patients to resume smoking. SC support has therefore been recommended for persons diagnosed with TB as a teachable moment [33] and an opportunity to reverse smoking-related lung health damage [9]. Studies conducted in SA have shown the efficacy of motivational interviewing skills in improving SC support [29]. Patients need to be educated about the fact that smoking, when experiencing an improvement in TB treatment outcome, is a way of reversing health gains; therefore, it is counterproductive to their general health and well-being. In addition, the participants mentioned the need for means of determining TB patients’ smoking status. The identification of smokers using biomarkers will not only be useful in identifying smokers but can also be used to monitor SC as has been used by researchers [29]. Nicotine addiction is a major contributory factor to failed quit-smoking attempts [28,43,44]; therefore, to help smokers who are addicted to quit smoking, pharmacotherapy is recommended in addition to counseling [28]. 

### 4.2. Personal Factors (Staff)

HS’s personal factors that may hinder SCIs include a lack of knowledge regarding SC aids and a lack of knowledge about the effects of smoking on TB treatment outcomes. While most HS were knowledgeable about the damage caused by smoking to lung health, they mentioned that it was beyond their professional roles to offer SCIs. However, there was no consensus among them on whether there are interactions between nicotine and TB drugs, which calls for further education and training on SCIs for hospital staff working with TB in-patients [25]. 4.3. Structural (Institutional) Factors

Structural-level barriers to SC support were also reported by HS, including a lack of SCIs and access to cigarettes within hospital premises, which facilitates continued smoking. Also, lack of time as exacerbated by staff shortages and prioritizing treatment adherence over other health goals were mentioned among the barriers to SC. Jeyashree and colleagues emphasized the need to study the effectiveness of pharmacological and behavioral SCIs in improving TB treatment outcomes [9]. Rigotti et al. concluded that high-intensity behavioral interventions promote SC in in-patients and recommended adding NRT to increase cessation rates [32]. As SC treatment has already been published in the Standard Treatment Guidelines for TB patients in SA [28], there is an opportunity to offer such support in public hospitals. Unfortunately, and as reiterated by this study’s participants and a WHO report [30], SC treatment is only available privately in SA. Access to cigarettes by patients within hospital premises is perceived to facilitate continued smoking behaviors, which undermine SC efforts among DR-TB in-patients in this study. The WHO, through the Framework Convention on Tobacco Control, advocates for a 100% smoke-free policy as the only proven way to protect people from the effects of smoking [30]. If incorporated into South African tobacco control law, such a policy can discourage smoking and eliminate access to cigarettes around hospital premises. On that note, Grable et al. (2023) propose that the government can address staff shortages by utilizing existing hospital staff and medical students, as demonstrated in the Rochester Model for tobacco dependence treatment in in-patients [45]. Similarly, other studies have also noted that the Opt-Out EHR-Based Service, which is an electronic health record system, triggers consultations for smoking patients, achieving a 50.4% clinician acceptance rate. Despite staff constraints limiting consultations to 17% of eligible patients, those receiving consultations had higher pharmacotherapy orders [46,47,48]. Moreover, research has shown that educational interventions such as training programs, which implement targeted educational sessions for junior doctors, has led to significant increases in referrals and prescriptions for nicotine replacement therapy (NRT), enhancing the capacity to support smoking cessation [49]. Nevertheless, despite these strategies showing promise, challenges remain, particularly in ensuring adequate staffing and resources to meet the demand for smoking cessation services in hospitals [50]. One of the main prevalent concerns is the shortage of HS, which leads to an overloaded work schedule and is currently a national phenomenon due to brain drain in the South African health sector [51]. The overloaded work schedules of HS, in turn, lead to a limited possibility of adding interventions like SC in the treatment plan for DR-TB in-patients [39]. HS in other countries, such as Pakistan, has raised such concerns on workloads in TB programs [24]. Organizational changes to accommodate SCIs by hospital staff are recommended to address this problem [23]. Also, it may be useful to explore dedicated smoking cessation clinics for DR-TB in-patients as suggested by one participant in this study. 

Nevertheless, various factors contribute to the limitation of the generalizability of the study, such as the fact that interviews with HS were generally short due to the limited time they had to spare to engage with the researcher. However, the length of the interviews did not limit the validity of the participants’ responses, as ensured by the triangulation of participants in this study. 

## 5. Conclusions

The health of DR-TB in-patients is further compromised when they smoke cigarettes. As SA makes efforts to curb TB incidence rates, DR-TB in-patients must be paid special attention since they are infected with strains of the TB-causing virus that can compromise the progress already recorded in TB treatment [23,39]. SCIs should therefore be included and made compulsory for DR-TB patients who smoke and for all TB patients in general. This will aid in assisting both the public and private healthcare sectors and professionals to tailor scientifically structured interventions that will improve overall TB treatment outcomes. 

## Figures and Tables

**Table 1 ijerph-21-01137-t001:** Breakdown of study participants.

Hospital Staff	Male	Female	Total
Doctor	2	1	3
Nurse	2	4	6
Nurse	1	3	4
Pharmacist	0	1	1
Pharmacy	0	1	1
Social Worker	0	2	2
Social Worker	0	1	1
Total number of participants			18

**Table 2 ijerph-21-01137-t002:** Barriers to smoking cessation support for drug-resistant tuberculosis in-patients.

Factors	Barriers
Personal factors (patients)	i. Non-disclosure of smoking status by patients. ii. Relapse due to improved TB conditions. iii. Addiction.
Personal factors (staff)	*i*. Lack of knowledge regarding smoking cessation aids. *ii*. Knowledge about the effects of smoking on TB recovery.
Structural (institutional) factors	*i*. Lack of smoking cessation interventions. *ii*. Emphasis on other health education matters. *iii*. Smoking cessation aids not considered in the government’s list of essential medicines. *iv*. Priority on treatment adherence only and lack of dedicated staff for SCIs. *v*. Access to cigarettes around hospital premises and non-compliance to smoking restrictions in hospital premises. *vi*. Lack of time and staff shortage.

TB: tuberculosis; SCIs: smoking cessation interventions.

## Data Availability

Data are contained within the article.

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
