# Peer review of "Perspectives of Hospital Staff on Barriers to Smoking Cessation Interventions among Drug-Resistant Tuberculosis Patients in a South African Management Hospital"

_ijerph, 2024, doi:10.3390/ijerph21091137_

Round 1
Reviewer 1 Report (Previous Reviewer 1)
Comments and Suggestions for Authors
NA
Author Response
Please see the attachment

Reviewer 2 Report (Previous Reviewer 3)
Comments and Suggestions for Authors
Thanks to the authors for the clarifications. The manuscript is now acceptable for publication.
Author Response
Please see the attachment

Reviewer 3 Report (New Reviewer)
Comments and Suggestions for Authors
A very well written and thoughtful paper. Sample size is small but 18 is reasonable for a study like this. Including selected pertinent comments verbatim enhances the paper considerably.
These two comments are absolutely sad and true for most of the world:
"“…but the main part of the hospital is so busy, so under-staffed, such large number of patients! So I don’t think it (smoking cessation intervention) will be a priority program in the near future. It will take a long time to come… because the staff is not enough” (Participant 11; Male, 311 Doctor)"
"“In the public sector, we don’t keep any nicotine replacement therapies … we’re working with limited resources so sometimes we don’t have sufficient finances to procure all the drugs that we need.........” (Participant 15; Female, Pharmacist).
Optional- but one thing that will enhance this article is a brief review of published papers or opinions from experts/ consensus guidelines or other respected sources, on how sometimes these challenges have been addressed.
Author Response
Please see the attachment

This manuscript is a resubmission of an earlier submission. The following is a list of the peer review reports and author responses from that submission.
Round 1
Reviewer 1 Report
Comments and Suggestions for Authors
1. Perspectives of Hospital Staff on Barriers to Smoking Cessation Interventions Among Drug-resistant Tuberculosis Patients in South African Management Hospitals.
As the study is conducted in only one hospital, it is better to write as ”……. in a South African Management Hospital”
2. As the data collection was conducted in 2015, was there any change in the policy? If yes whether it is addressed and updated?
3. Was a deductive or inductive method used while coding the interview transcripts?
4. In Line number 140, as there is only one nurse manager interviewed, it is disclosing the identity of the person.
5. Write the statement of the participants in double-quotes.
6. Overall paper is well written.
Reviewer 2 Report
Comments and Suggestions for Authors
The paper presented an interesting topic; however, there are significant issues that prevent it from being suitable for publication at this time. Firstly, the sample size was extremely low, which severely limits the ability to generalize the findings to a broader population. This small sample size raises concerns about the statistical power of the study and the reliability of the results.
Additionally, the paper is poorly written, with numerous grammatical and structural issues that hinder the clarity and flow of the content. Specific areas needing improvement include the introduction, which lacks a clear rationale for the study, and the discussion, which fails to adequately relate the findings to existing literature.
Furthermore, the methodology section is insufficiently detailed, making it difficult to assess the validity of the approach. More information is needed on the selection criteria for participants, the procedures followed, and the statistical analyses performed.
Due to these substantial deficiencies, I recommend that the paper should not be accepted for publication. Extensive revisions are required, including increasing the sample size, enhancing the clarity and coherence of the writing, and providing more detailed methodological and contextual information.
Comments on the Quality of English LanguageThe paper presented an interesting topic; however, there are significant issues that prevent it from being suitable for publication at this time. Firstly, the sample size was extremely low, which severely limits the ability to generalize the findings to a broader population. This small sample size raises concerns about the statistical power of the study and the reliability of the results.
Additionally, the paper is poorly written, with numerous grammatical and structural issues that hinder the clarity and flow of the content. Specific areas needing improvement include the introduction, which lacks a clear rationale for the study, and the discussion, which fails to adequately relate the findings to existing literature.
Furthermore, the methodology section is insufficiently detailed, making it difficult to assess the validity of the approach. More information is needed on the selection criteria for participants, the procedures followed, and the statistical analyses performed.
Due to these substantial deficiencies, I recommend that the paper should not be accepted for publication. Extensive revisions are required, including increasing the sample size, enhancing the clarity and coherence of the writing, and providing more detailed methodological and contextual information.
Reviewer 3 Report
Comments and Suggestions for Authors
This is a good qualitative study assessing the input of different hospital staff regarding smoking cessation in patients with drug-resistant TB. A few issues need to be addressed:
1. Abstract: The last sentence needs evidence. Did you run any statistics to support our claim of improvement? If not, then it's better to soften the language. For example, instead of "were found to", you may use "were assumed to..."
2. Keywords: I suggest adding "tuberculosis"
3. Introduction (line 64): References 21 and 22 are more than 10 years old. Could you find a more recent reference?
4. Methods (Study Participants and Ethical Approval): What were the inclusion criteria? And how did you approach the 18 participants to take part in the study? What type of sampling did you use?
5. Methods (line 100): I suggest removing the first sentence "This study was part of the doctoral research of the first author."
6. Table 1 should be moved to the results section.
7. Data collection: Did you obtain written or verbal consent to participate in the study?
8. Data collection: Did you run a pilot phase? If so, please add the details and whether or not you edited the questions based on the feedback from the pilot phase.
9. Line 121: Add the name of the software developer and their city and country.
10. Results: The results section should begin with the characteristics of the participants. Thus, describe Table 1 and then insert it here before describing Table 2.
11. Table 2: Add the abbreviations under the table.
12. When typing the responses of the participants in their words, it's better to add quotation marks and perhaps italicize the text, so it can be distinguished from the main text of the manuscript.
13. Lines 186-191: Are these aids provided by the hospital to the smoking patients? Is it a standard practice in your institution to offer these aids to all smoking patients? Please provide some clarification in this very short section. Alternatively, this addition could be added to the respective section in the discussion.
14. Lines 366-368: How could this be a limitation when it's the aim of the study? I suggest deleting it.
15. Line 375: TB is not caused by a virus. It is a mycobacterium. Please correct.
Round 2
Reviewer 3 Report
Comments and Suggestions for Authors
Thank you to the authors for addressing the comments and suggestions. However, it seems like three comments (#8, 10 and 13) were not addressed nor rebutted in the authors' response file. The authors highlighted the comments in yellow, which wasn't clear to me what did this indicate. If you have a justification for not addressing these comments, then please state your justification.
